# Examining the Predictors of Mental Ill Health in Esport Competitors

**DOI:** 10.3390/healthcare10040626

**Published:** 2022-03-26

**Authors:** Matthew Smith, Benjamin Sharpe, Atheeshaan Arumuham, Phil Birch

**Affiliations:** 1School of Sport, Health and Community, Faculty of Health and Wellbeing, University of Winchester, Winchester SO22 4NR, UK; 2Institute of Education, Social and Life Sciences, University of Chichester, Chichester PO19 6PE, UK; b.sharpe@chi.ac.uk; 3Department of Psychosis Studies, Institute of Psychiatry, Psychology and Neuroscience, King’s College London, London SE5 8AF, UK; atheeshaan.arumuham@kcl.ac.uk; 4Institute of Sport, Nursing and Allied Health, University of Chichester, Chichester PO19 6PE, UK; p.birch@chi.ac.uk

**Keywords:** stressors, mental ill health, hierarchical regression, esports

## Abstract

Few research studies have examined the predictors of mental ill health in esports. This study addresses that gap by investigating stressors, sleep, burnout, social phobia anxiety and mental ill health in esport athletes. An online survey was disseminated to competitive student esport athletes (*n* = 313) residing in the UK. The survey included measures of stressors resulting from competing in esports, sleep quality, burnout, and social phobia, as well as outcome measures of mental ill health. Hierarchical regression analyses examined these relationships. All the hypotheses were supported, with stressors significantly predicting sleep quality, burnout, and social phobia anxiety, and stressors, sleep quality, burnout, and social phobia anxiety were all significant positive predictors of mental ill health. The strength of these predictions varied, for example, the daytime dysfunction subscale of sleep was a strong predictor of all outcome variables; two subscales of burnout, reduced sense of accomplishment and exhaustion significantly predicted each of the three mental ill health outcome variables, and two subscales of social phobia anxiety, fear and avoidance, significantly predicted mental ill health. Our study has important implications for player health in esports, highlighting interventions that could target specific aspects of stress, sleep, burnout, and social phobia anxiety to improve the mental health of those who compete in esports.

## 1. Introduction

The incidence of mental ill health has shown to be highly prevalent in esport athletes [1], which is a comparable level to other professional sports (e.g., football [2]). Mental health can be defined as “a state of wellbeing in which every individual realizes his or her own potential, can cope with the normal stresses of life, can work productively and fruitfully, and is able to make a contribution to his or her community” (p. 231, [3]). Mental ill health refers to a negative state of wellbeing, and severe mental ill health refers to diagnosable disorders such as depression or anxiety. Improving the mental health of athletes is now a priority for many sporting organizations [4], and researchers [5,6] have directly highlighted the need for quality healthcare provision for esport athletes and higher quality data to underpin prevention strategies. However, limited research has directly investigated the impact of stressors on the mental health of esport athletes, which is needed to provide an evidence-base to underpin strategies to improve their mental health.

In any competitive context, it is important to understand the unique stressors that are detrimental to mental health, including sport-related stressors [7]. Esport athletes face specific demands that vary to those faced by traditional sports athletes, for example, the need to use skilled fine motor co-ordination while facing a high cognitive workload that includes attention, information processing and visuo-spatial skills [8]. In addition, whilst most sports have on- and off-seasons, esport athletes can compete in multiple tournaments throughout the year, with one study identifying that elite esports athletes train an average 5.28 h a day year-round [9]. Such intense practice schedules have been associated with esport athletes experiencing social anxiety, depression, and burnout [8]. A large body of research in sport psychology literature has focused on the environmental demands in competitive sporting contexts, for example, stressors faced by elite cricket captains [10] and elite sport coaches [11]. In esports, a systematic review has highlighted the psychological and physiological stress experienced in esport settings, including stress generated by persistent motivated performance, and performance anxiety [12]. Studies examining the physiological response to playing esports have found athletes experience a significant increase in cardiac output, at times reaching heart rates of 120 beats per minute [13]. Researchers have also used qualitative methods to identify stressors faced by esport athletes. In a sample of League of Legends competitors, stressors included harassment by others, and negative communication during performance [14]. Research with a sample of Counter-Strike: Global Offensive (CS:GO) competitors identified a range of internal (e.g., interpersonal issues within teams) and external (e.g., performing in front of an audience and doing media interviews) stressors [15]. While this research has identified stressors faced by esport athletes, research is needed to directly investigate the impact of stressors on the mental health of esport athletes [12].

In further understanding the predictors of mental ill health in esport athletes, sleep, burnout, and social phobia anxiety are three further variables of interest in the current study. First, researchers have highlighted that sleep can impact negatively on the cognitive functioning of athletes, which can in turn have a negative impact on performance [16]. When considering mental ill health, sleep problems and poor sleep hygiene have been seen to predict low mood in a sample of elite multi-sport athletes competing at the World Student Games [17]. In addition, in a sample of retired elite soccer players, when adverse life events had occurred, sleep disturbance was seen to lead to increased reporting of mental distress in retired elite soccer players [18]. More recently, researchers have identified the risk factors for poor sleep in esport athletes, with negative physiological and psychological impacts for esport athletes seen to be associated with poor sleep quality (see the work of Bonnar and colleagues for a review [19]). For example, research has highlighted the prevalence of competitive late night playing sessions and that esport athletes reported significantly lower sleep quality and higher depression scores when compared to a non-athlete sample [20]. Furthermore, esport athletes were seen to report various problems with sleep, and this correlated with higher levels of severe depression scores [21]. The preliminary evidence gleaned from these studies suggests esport athletes may be predisposed to sleep problems, which may lead to mental ill health. Further research is therefore needed to investigate the predictive relationships between sleep and mental ill health in esport athletes. 

A second variable of interest in this present study is burnout and its association with stressors and mental ill health. Burnout has been conceptualized as feelings of exhaustion, resulting from chronic workplace stress that has not been successfully managed [22]. Research has consistently highlighted the relationship between stress and burnout [23], with stressors faced by junior doctors during a residency programme impacting on burnout [24]. Chronic stress has been shown to be a significant predictor of both burnout and depression in a sample of junior athletes [25] and in a longitudinal study with a sample of Finnish forestry workers, elements of the working environment created greater levels of burnout, which in turn was argued to lead to mental ill health [26]. Further research in sport has identified that symptoms of burnout have been shown to be significant contributors to diminished psychological wellbeing [27]. Despite esport athletes reporting problems with burnout in the media [28], there is an absence of research explicitly examining factors contributing to the prevalence of burnout, and further research is needed to investigate the predictive relationships between stressors, burnout, and mental ill health in esport athletes.

Third, we consider social phobia anxiety, which is a disorder that encompasses a distorted and maladaptive view of oneself [29], whereby socially phobic individuals are concerned about characteristics of self that they consider as being deficient when compared to societal expectations, and fear the consequences that may occur (i.e., negative evaluation, rejection, embarrassment, loss of societal status) if those self-attributes are exposed to scrutiny by critical others. Researchers have found social phobia anxiety to predict self-reported sensory-processing sensitivity, social/childhood fears, and behavioral inhibition [30]. Furthermore, students who experienced social phobia anxiety reported significantly lower levels of quality of life (i.e., general health, vitality, social functioning, role functioning-emotional) and mental health [31]. Social phobia anxiety has also been shown to positively relate to symptoms of depression in recreational video gaming [32] and to internet gaming disorder symptoms [33]. Despite preliminary evidence suggesting that esport athletes find it stressful to embark in interactions with new environments and people [15], research is needed to examine whether social phobia anxiety predicts mental ill health in esport athletes.

In summary, there is a paucity of research examining mental health in esports. In this study we aimed to advance the literature by examining the predictors of mental ill health in esport athletes. It is hoped that the findings from this study elucidate the risk factors underpinning mental ill health, and in turn, provide practitioners with underpinning evidence-based healthcare provision to support esport athletes. Based upon the current literature, we formulated the following five hypotheses; (i) stressors will significantly and negatively predict quality of sleep, and will significantly and positively predict burnout and social phobia anxiety; (ii) stressors will significantly and positively predict mental ill health; (iii) quality of sleep will significantly and positively predict mental ill health; (iv) burnout will significantly and positively predict mental ill health; (v) social phobia anxiety will significantly and positively predict mental ill health.

## 2. Materials and Methods

### 2.1. Participants and Procedure

A total of 313 current competitive esport athletes (*M_age_* = 19.8, *SD* = 2.0 years), with varying durations of videogame (*M* = 12.1, *SD* = 4.1 years) and esport experience (*M* = 2.8, *SD* = 4.9 years), participated in the study. The sample consisted of 30 females and 283 males, from three esports which included CS: GO (*n* = 165), Valorant (*n* = 68), and Rainbow Six Siege (*n* = 80), which are all multi-player, competitive, first-person shooter games, with the aim of out-performing the opposition team. G*Power 3.1.9.4 software was used to perform an a priori calculation of sample size [34]. With a power (1 − β) of 0.95, two-tailed α of 0.05 and the set of predictors, 234 participants were required to detect a medium effect (f2 = 0.15).

Over a period of three months, cross-sectional data was collected using Qualtrics (London, UK). Ethical approval for the study protocol was awarded by the lead* institution. Criteria-based sampling was used via collaborator networks, social media and partnership with a university-based esport tournament organization (Nuel). Participants (≥18 years), comprised of student-level esport competitors from CS: GO, Valorant, and Rainbow Six Siege. Once informed consent was provided, participants provided demographic information and completed a battery of questionnaires, accessed via a unique study link, relating to stressors faced when competing in esports, sleep (Pittsburgh Sleep Quality Index [PSQI]) [35], burnout (Athlete Burnout Questionnaire [ABQ]) [36], social phobia anxiety (Social Phobia Inventory (SPIN]) [37], general anxiety and depression (General Health Questionnaire—short form [GHQ-12] [38], and more severe depressive symptoms (Patient Health Questionnaire [PHQ-9]) [39], and psychological distress (Distress Screener) [40].

### 2.2. Measures

#### 2.2.1. Stressors Measure

We developed a scale to assess the stressors pertinent to our sample of esport competitors. First, we developed a pool of 57 stressors using previous literature from esports [12,14] and other sporting contexts (e.g., [41]). This pool was scrutinized by an external panel of key stakeholders from the esport community using a 5-point rating scale (ranging from 1 ‘not stressful’ to 5 ‘extremely stressful’). This led to the removal of 24 irrelevant stressors, which resulted in 33 stressors (e.g., “aggressive communication from in-game leader”) being taken forward for analyses. To organize the stressors into categories, principal component analysis (PCA) was applied. Orthogonal varimax rotation was applied to the component matrix, which seeks to increase the variances of the factor loadings (i.e., factor loadings above 0.4 were considered significant). Seven factors were extracted from the PCA with eigenvalues greater than one. These seven factors were labelled, (1) teammate interactions, (2) personal concerns, (3) teammate concerns, (4) game-specific worry, (5) IGL-specific interactions, (6) game-specific uncertainty, and (7) in-game pressure. See Appendix A for more information on how the stressor categories were calculated, and which stressors were in each category.

#### 2.2.2. Pittsburgh Sleep Quality Index (PSQI)

Sleep quality was measured using the PSQI [35], which is a 19-item self-report measurement and groups scores into seven components: subjective sleep quality, sleep latency, sleep duration, sleep efficiency, sleep disturbances, daytime dysfunction (i.e., general enthusiasm), and use of sleep medication. This scale has previously demonstrated good internal consistency (see [42] for a review). Cronbach’s Alpha for the scale in this study was good (α = 0.87).

#### 2.2.3. Athlete Burnout Questionnaire (ABQ)

The ABQ [36] is a 15-item instrument which measures self-reported levels of burnout. The ABQ has three subscales related to (i) emotional/physical exhaustion, (ii) reduced sense of accomplishment and (iii) sport devaluation. Questions were revised to be specific to that of the target audience (e.g., “I don’t care as much about my sport performance as I used to do” was revised to “I don’t care as much about my esport performance as I used to”). This scale has previously demonstrated good internal consistency [43]. Cronbach’s Alpha for the scale in this study was acceptable, with emotional/physical exhaustion (α = 0.88), reduced sense of accomplishment (α = 0.71), and sport devaluation (α = 0.80).

#### 2.2.4. Social Phobia Inventory (SPIN)

Social phobia anxiety was measured using the SPIN [37] which is a 17-item measure with three subscales related to (i) fear, (ii) avoidance, and (iii) physiological distress. Previous literature has demonstrated good internal consistency for the SPIN [44]. The internal consistency of the SPIN in this study was good (α = 0.94), including the fear subscale (α = 0.91), avoidance (α = 0.80), and physiological distress (α = 0.89).

#### 2.2.5. General Health Questionnaire—Short Form (GHQ-12)

This study used the GHQ-12 [38], a 12-item instrument designed to assess psychological signs of anxiety or depression experienced by the participants in the previous four weeks. Previous literature has demonstrated good internal consistency of the GHQ-12 [45,46]. The internal consistency of the GHQ-12 in this study was acceptable (α = 0.73).

#### 2.2.6. Patient Health Questionnaire (PHQ-9)

The PHQ-9 [39] is a nine-item instrument to assess severe depressive symptoms. Previous literature has demonstrated good internal consistency of the PHQ-9 [47]. The internal consistency of the PHQ-9 in this study was good (α = 0.87).

#### 2.2.7. Distress Screener

A three-item distress screener [40], based on Four-Dimensional Symptom Questionnaire (4DSQ) [40], was used to assess psychological symptoms of distress experienced in the previous 4 weeks. Previous literature has demonstrated good internal consistency for this scale [48,49], and in the current study, the internal consistency of the distress screener was good (α = 0.80).

### 2.3. Data Analysis Strategy

Pearson’s correlation analyses were conducted to examine the association between the tested variables. For sleep, burnout, and social phobia anxiety, only subscale scores were used as these, allowing the examination of the unique contributions in the regression analyses. Table 1 displays correlations between all predictor and outcome variables. All multicollinearity (i.e., Durbin–Watson test, Tolerance, Variance Inflation Factor), multivariate normality (i.e., Mahalanobis distance), and outlier checks met relevant analysis assumptions. A series of hierarchical multiple regression analyses were conducted to determine whether independent variables are statistically significant predictors of a dependent variable and provide the respective contribution of independent variables in said prediction. The alpha value was set at 0.05 for all analyses. The relative strength of the unique predictions of individual variables (i.e., standardized beta coefficients) were interpreted using the recommendations of Cohen (1988), who defined values near 0.02 as small, near 0.15 as medium, and above 0.35 as large.

## 3. Results

Descriptive statistics of all the outcome variables can be seen in Appendix A. Correlation analyses revealed multiple significant findings. A series of hierarchical multiple regression analyses further revealed that stressors significantly predict sleep, burnout, and social phobia anxiety (Hypothesis 1—see Table 2), and the predictors of mental ill health which included stressors (Hypothesis 2–5—see Table 3). These findings are considered more fully in the following discussion section.

## 4. Discussion

### 4.1. Stressors as Predictors of Sleep, Burnout, and Social Phobia Anxiety

Findings provide support for the first hypothesis that stressors would negatively predict quality of sleep and positively predict burnout and social phobia anxiety. When examining the relationship between stressors and sleep, stressors significantly and positively predicted some sleep variables, including a small positive relationship with sleep duration (8.9% variance explained), but the stressors also had small positive and significant relationships with maladaptive elements of sleep, including sleep disturbance (9.3%) and daytime dysfunction (14.7%). These findings support propositions that risk factors associated with esports can be detrimental to sleep quality [19]. In terms of the relationships between stressors and burnout, significant medium strength predictions were seen with two of the subcomponents of burnout, which supports previous research that has found associations between stressors and burnout [24]. Findings showed that stressors significantly and negatively predicted 15.0% of the variance of reduced sense of accomplishment and 18.4% of the variance of exhaustion. Four categories of stressors each significantly and positively predicted these two subcomponents of burnout, category 1 (teammate interactions), 2 (personal concerns), 3 (teammate concerns), and 6 (game-specific uncertainty). The relationship between stressors and burnout was only partially supported as stressors did not significantly predict the sport devaluation component of burnout. Finally, stressors significantly and positively predicted each of the three subcomponents of social phobia anxiety, predicting 26.6% of the variance of fear, 15.6% of the variance of avoidance, and 15.5% of the variance of physiological symptoms. Various categories of stressors were significant predictors of social phobia anxiety, but category 2 (personal concerns) and category 7 (in-game pressure) were the strongest predictors of all three subcomponents. Overall, our results offer strong support for the first hypothesis that stressors experienced by esport athletes predict sleep quality, burnout, and social phobia.

### 4.2. Stressors Predicting Variables of Mental Ill Health

The second hypothesis, that stressors would positively predict mental ill health was also supported. Findings showed that stressors significantly and positively predicted mental ill health, with the seven stressor categories predicting 26.4% of the variance of general signs of anxiety and depression (as measured by the GHQ-12). There was also a medium strength predictive relationship (16.3%) between stressors and the psychological distress (as measured by the distress screener), and a small predictive relationship (12.7%) between stressors and more severe mental ill health (i.e., depressive symptoms, as measured by the PHQ-9). When considering the seven categories of stressors, three categories all were significant predictors of the three variables of mental ill health, category 2 (personal concerns), category 6 (game-specific uncertainty), and category 7 (in-game pressure). Foskett and Longstaff [40] theorized that the stress of competing at an elite level may predict mental ill health, and found that career dissatisfaction predicted signs of anxiety, depression, and distress within their sample of athletes. Our findings support and extend this research [40] of by providing evidence that different stressors (caused by competing in esports) can predict mental ill health. Findings also extend previous work that has examined stressors in esports [14,15] as to our knowledge, this is the first study to directly provide evidence that stressors can predict mental ill health in esport athletes.

### 4.3. Sleep Predicting Variables of Mental Ill Health

Findings provide strong support for the third hypothesis, with sleep being a strong predictor of the respective variables of mental ill health. Sleep significantly predicted 33.7% of the variance of general signs of anxiety and depression, and 49.2% of the variance of more severe depression, and 41.4% of the variance of psychological distress. The findings provide further evidence of the impact of sleep on mental ill health and support the work of Lee and colleagues [20,21]. These researchers found that lower sleep quality might explain more severe issues of mental ill health, with esport athletes reporting worse sleep quality and greater levels of depression (including clinical symptoms of depression [20]), with sleep problems being strongly correlated to depression [21]. Our findings support and extend this research by showing a prediction between sleep quality and issues with mental ill health, as well as elucidating the predictive contribution of different sleep components. We found that for each outcome variable of mental ill health, daytime dysfunction strongly predicted general anxiety and depression (β = 0.51), and strongly predicted severe depression (β = 0.61). Findings support previous research that has found a specific association between daytime disturbance and mental ill health in student samples [50], and research that has highlighted daytime dysfunction as a key problem for esport athletes, regardless of playing level [51]. In the general population, disrupted sleep cycles are associated with impaired cognitive function, increased risk of cardio-metabolic diseases, and mood disturbances which include suicidal ideation and engaging in risky behaviors [52]. Findings therefore suggest that improving sleep quality, and particularly addressing daytime dysfunction, could be key in improving the mental and physical health of esport athletes.

### 4.4. Burnout Predicting Variables of Mental Ill Health

Findings also provide strong support for the fourth hypothesis, with burnout significantly predicting both mental ill health variables. Burnout significantly and positively predicted 23.4% of the variance of general signs of anxiety and depression, 29.2% of the variance of more severe depression, and 17.2% of the variance of psychological distress. These results support previous findings [26] that stress in the work environment led to increased burnout, particularly exhaustion, which then led to issues of mental ill health such as depression. Findings also support and extend findings in that burnout was significantly associated with depression [25]. Our differentiated analysis revealed two of the three subscales of burnout, reduced sense of accomplishment and exhaustion, to be large significant predictors each of the mental ill health outcome variables. Specifically, exhaustion was the strongest predictor of general anxiety and depression (β = 0.45), more severe depression (β = 0.52), and psychological distress (β = 0.40). The emotional and physical exhaustion athletes experience might be linked to intense demands of training and competition when playing sport [36]. This exhaustion is likely to be present for esport athletes, as research has evidenced how they can average 60 h of play per week [53]. Burnout has been suggested to be a major problem facing esport athletes, which in turn, could lead to premature retirement, and problems with mental ill health in the post-career phase [54]. Therefore, our findings reinforce how burnout appears to be important risk factor when considering causes of mental ill health in esport athletes.

### 4.5. Social Phobia Anxiety Predicting Variables of Mental ILL Health

Finally, findings also provide strong support for the fifth hypothesis, with social phobia anxiety being a medium strength predictor of each of our variables of mental ill health. Social phobia anxiety significantly and positively predicted 33.2% of the variance of general signs of anxiety and depression, 30.8% of the variance of more severe depression, and 32.5% of the variance of psychological distress. Our findings extend research that found positive associations between social phobia anxiety and mental ill health in online video players [32] and internet gaming disorder symptoms [33], as our results reveal two of the three subscales of social phobia anxiety (fear and avoidance) significantly predicted mental ill health outcome variables. With social phobia anxiety seen to relate to depressive disorders in a 2 to 5-year follow-up period in adolescents and young adults [55], our preliminary findings have meaningful implications for the prediction of long-term mental health in esport athletes.

### 4.6. Limitations

A strength of this study was the large sample of competitive esport athletes that allowed us to investigate the predictive relationships of stressors, key variables, and mental ill health. We acknowledge that our sample comprised student players, and future research should use a more elite sample, as researchers have identified athletes who compete at the elite level are exposed to unique factors that could impact mental ill health [27,56]. Furthermore, we considered the subscales of the variables which allowed us to provide more differentiated results, for example, which specific elements of sleep and burnout predicted mental ill health. Indeed, given the importance of daytime dysfunction in the findings of the current study, future research might use other instruments to measure sleep such as the Epsworth sleepiness scale [57], which more directly measures daytime dysfunction. A further limitation of the current study was the cross-sectional nature of the data collection, which makes it challenging to infer causality and specify the order the predictive relationships coherently. For example, our findings show burnout leads to mental ill health, but it is possible that this direction could be reversed, with previous research revealing support for cross-paths in both directions when investigating associations between burnout and depression [25]. Future research might use a longitudinal methodology, which would allow a more robust interpretation of such predictive relationships and may strengthen our ability to interpret findings.

### 4.7. Practical Implications

Our results highlight sleep as a key variable that is linked with mental ill health. Thus, interventions are needed to improve the quality of sleep of esport athletes to improve their mental health. When considering the efficacy of interventions, practitioners aiming to remedy problems with poor sleep quality are directed to the work of Bonnar et al. [19] which provides a review of the evidence supporting different sleep interventions. Our results also highlight the predictive relationships between stressors, burnout, and mental ill health. To address/reduce burnout, interventions should target both the structure of the sport, considering the organizational or environmental demands, and individuals, giving esport athletes specific strategies to deal with these demands [23]. Findings provide preliminary evidence to develop strategies that target pertinent stressors, as we have identified the specific categories of stressors associated with burnout (i.e., teammate interactions, personal concerns, teammate concerns, and game-specific uncertainty). Stressors predicted social phobia anxiety, and, in turn, social phobia anxiety predicted mental ill health. Findings provide initial evidence to address the specific stressors (personal concerns and in-game pressure) that are associated with social phobia anxiety.

## 5. Conclusions

The findings of the current study identified specific categories of stressors which predicted sleep, burnout, social phobia anxiety, and mental ill health. In addition, we identified the specific elements of sleep, burnout and social phobia anxiety that predicted mental ill health. This study provides preliminary evidence to elucidate the key risk factors underpinning mental ill health in university-based esport athletes. It is hoped that these findings underpin evidence-based healthcare provision to support mental health in esports.

## Figures and Tables

**Table 1 healthcare-10-00626-t001:** Correlation coefficients of analysed variables.

Variable	8.	9.	10.	11.	12.	13.	14.	15.	16.	17.	18.	19.	20.	21.	22.	23.
1. PCA1	0.052	0.07	0.091	0.103	0.059	0.088	0.004	0.204 ***	0.152 **	0.027	−0.093	−0.058	−0.13 *	0.11	0.075	0.033
2. PCA2	0.142 *	0.103	0.136 *	0.078	0.178 **	0.001	0.218 **	0.214 **	0.187 ***	0.052	0.299 ***	0.22 ***	0.216 ***	0.308 ***	0.232 ***	0.24 ***
3. PCA3	0.013	0.052	0.211 ***	0.14 *	0.121 *	0.051	0.104	0.16 **	0.21 ***	−0.027	0.11	0.136 *	0.028	0.207 ***	0.079	0.059
4. PCA4	0.002	0.033	0.029	−0.044	0.071	−0.02	0.032	0.074	0.095	0.011	0.069	0.028	0.058	0.104	0.032	0.041
5. PCA5	−0.014	−0.032	0.06	0.055	0.014	0.136 *	0.058	0.026	0.121 *	0.112 *	0.231 ***	0.095	0.196 ***	0.141 *	0.09	0.113 *
6. PCA6	0.085	0.083	0.074	0.178 *	0.166 **	0.134 *	0.277 ***	0.171 **	0.242 ***	0.154 **	0.143 *	0.165 **	0.019	0.252 ***	0.199 ***	0.23 ***
7. PCA7	−0.011	0.041	0.091	0.01	0.099	0.081	0.085	0.033	0.01	−0.05	0.277 **	0.221 ***	0.22 ***	0.141 ***	0.111 *	0.181 **
8. SQ	—	0.416 ***	0.269 ***	0.188 ***	0.261 ***	−0.037	0.276 ***	0.096	0.112 *	0.005	0.13 *	0.146 **	0.049	0.18 **	0.383 ***	0.303 ***
9. SL		—	0.257 ***	0.263 ***	0.325 ***	0.241 ***	0.285 ***	0.17 **	0.283 ***	0.142 *	0.189 ***	0.181 **	0.079	0.275 ***	0.418 ***	0.319 ***
10. SDur			—	0.528 ***	0.168 **	0.132 *	0.284 ***	0.248 ***	0.255 ***	0.176 **	0.246 ***	0.28 ***	0.201 ***	0.306 ***	0.297 ***	0.229 ***
11. SE				—	0.238 ***	0.187 ***	0.247 ***	0.228 ***	0.274 ***	0.251 ***	0.174 **	0.254 ***	0.119 *	0.256 ***	0.237 ***	0.196 ***
12. SDis					—	0.152 **	0.303 ***	0.269 ***	0.312 ***	0.245 ***	0.214 ***	0.324 ***	0.142 *	0.335 ***	0.411 ***	0.321 ***
13. SMed						—	0.222 ***	0.197 ***	0.262 ***	0.151 **	0.231 ***	0.235 ***	0.149 **	0.245 ***	0.213 ***	0.162 **
14. DDys							—	0.315 ***	0.382 ***	0.266 ***	0.454 ***	0.491 ***	0.294 ***	0.514 ***	0.614 ***	0.611 ***
15. B-RA								—	0.53 ***	0.379 ***	0.312 ***	0.326 ***	0.207 ***	0.38 ***	0.41 ***	0.313 ***
16. B-E									—	0.58 ***	0.319 ***	0.396 ***	0.232 ***	0.452 ***	0.516 ***	0.396 ***
17. B-D										—	0.136 *	0.239 ***	0.114 *	0.236 ***	0.329 ***	0.252 ***
18. SP-F											—	0.751 ***	0.65 ***	0.547 ***	0.51 ***	0.549 ***
19. SP-A												—	0.531 ***	0.53 ***	0.527 ***	0.513 ***
20. SP-PS													—	0.351 ***	0.358 ***	0.38 ***
21. GHQ-12														—	0.585 ***	0.67 ***
22. PHQ-9															—	0.671 ***
23. DS																—

*Note.* PCA1-7 = stressor categories; SDur = sleep duration; SQ = sleep quality; SL = sleep latency; SE = sleep efficiency; SDis = sleep disturbance; DDys = daytime dysfunction; SMed = sleep medication; B-E = burnout exhaustion; B-RA = reduced sense of accomplishment; B-D = sports devaluation; SP-F = social phobia anxiety fear; SP-A = social phobia anxiety avoidance; SP-PS = social phobia anxiety physiological symptoms; DS = distress screener; * *p* < 0.05; ** *p* < 0.01; *** *p* < 0.001.

**Table 2 healthcare-10-00626-t002:** Hierarchical regression analyses: Effects of stressors on outcome variables.

DV	Step	*R*²	β	DV	Step	*R*²	β	DV	Step	*R*²	β	DV	Step	*R*²	β
SDur	1. PCA3	0.10 *	0.21 *	SQ	1. PCA2	0.02 *	0.14 *	SL	1. PCA1	0.03	0.07	SE	1. PCA6	0.03 *	0.18 *
	2. PCA2	0.06 *	0.14 *		2. PCA1	0.03	0.05		PCA2		0.10		2. PCA3	0.05 *	0.14 *
	3. PCA1	0.09 *	0.09		PCA3		−0.01		PCA3		0.05		3. PCA1	0.07 *	0.10
	PCA4		0.03		PCA4		0		PCA4		0.03		PCA2		0.08
	PCA5		0.06		PCA5		−0.01		PCA5		−0.03		PCA4		−0.04
	PCA6		0.07		PCA6		0.09		PCA6		0.08		PCA5		0.06
	PCA7		0.09		PCA7		−0.01		PCA7		0.04		PCA7		0.01
SDis	1. PCA2	0.03 *	0.18 *	DDys	1. PCA6	0.08 *	0.28 *	SMed	1. PCA5	0.02 *	0.14 *	B-E	1. PCA6	0.06 *	0.24 *
	2. PCA6	0.06 *	0.17 *		2. PCA2	0.12 *	0.22 *		2. PCA6	0.04 *	0.14 *		2. PCA3	0.10 *	0.21 *
	3. PCA3	0.07 *	0.12 *		3. PCA1	0.15 *	0		3. PCA1	0.05 *	0.09		3. PCA2	0.14 *	0.19 *
	4. PCA1	0.09 *	0.06		PCA3		0.10		PCA2		0		4. PCA1	0.16 *	0.15 *
	PCA4		0.07		PCA4		0.03		PCA3		0.05		5. PCA5	0.18 *	0.12 *
	PCA5		0.01		PCA5		0.06		PCA4		−0.02		6. PCA4	0.18 *	0.10
	PCA7		0.01		PCA7		0.09		PCA7		0.08		PCA7		0.01
B-RA	1. PCA1	0.04 *	0.20 *	B-D	1. PCA6	0.02 *	0.05 *	SP-F	1. PCA2	0.09 *	0.30 *	SP-A	1. PCA7	0.05 *	0.22 *
	2. PCA2	0.09 *	0.21 *		2. PCA5	0.04 *	0.12 *		2. PCA7	0.17 *	0.28 *		2. PCA2	0.10 *	0.22 *
	3. PCA6	0.12 *	0.17 *		3. PCA1	0.04	0.35		3. PCA5	0.22 *	0.23 *		3. PCA6	0.12 *	0.17 *
	4. PCA3	0.14 *	0.16 *		PCA2		0.05		4. PCA6	0.24 *	0.14 *		4. PCA3	0.14 *	0.14 *
	5. PCA4	0.15 *	0.07		PCA3		−0.03		5. PCA1	0.27 *	−0.09		5. PCA1	0.16 *	−0.06
	PCA5		0.03		PCA4		0.01		PCA3		0.11 *		PCA4		0.03
	PCA7		0.03		PCA7		−0.05		PCA4		0.07		PCA5		0.10
SP-PS	1. PCA7	0.05 *	0.22 *	GHQ-12	1. PCA2	0.10 *	0.31 *	DS	1. PCA2	0.06 *	0.24 *	PHQ-9	1. PCA2	0.05 *	0.23 *
	2. PCA2	0.10 *	0.22 *		2. PCA6	0.16 *	0.25 *		2. PCA6	0.11 *	0.23 *		2. PCA6	0.09 *	0.20 *
	3. PCA5	0.13 *	0.20 *		3. PCA3	0.20 *	0.21 *		3. PCA7	0.14 *	0.18 *		3. PCA7	0.11 *	0.11 *
	4. PCA1	0.15 *	−0.13 *		4. PCA5	0.22 *	0.14 *		4. PCA5	0.16 *	0.11 *		4. PCA1	0.13 *	0.08
	5. PCA3	0.16 *	0.03		5. PCA7	0.24 *	0.14 *		5. PCA1	0.16 *	0.03		PCA3		0.08
	PCA4		0.06		6. PCA1	0.26 *	0.11 *		PCA3		0.06		PCA4		0.03
	PCA6		0.02		PCA4		0.10 *		PCA4		0.04		PCA5		0.09

DV = dependent variable; PCA1-7 = stressor categories; SDur = sleep duration; SQ = sleep quality; SL = sleep latency; SE = sleep efficiency; SDis = sleep disturbance; DDys = daytime dysfunction; SMed = sleep medication; B-E = burnout exhaustion; B-RA = reduced sense of accomplishment; B-D = sports devaluation; SP-F = social phobia anxiety fear; SP-A = social phobia anxiety avoidance; SP-PS = social phobia anxiety physiological symptoms; DS = distress screener; * *p* < 0.05.

**Table 3 healthcare-10-00626-t003:** Hierarchical regression analyses: Effects of variables on mental ill health.

DV	Step	*R*²	β	DV	Step	*R*²	β	DV	Step	*R*²	β
PHQ-9	1. DDys	0.38 *	0.61 *	GHQ-12	1. DDys	0.26 *	0.51 *	DS	1. DDys	0.37 *	0.61 *
	2. SL	0.44 *	0.27 *		2. SDis	0.30 *	0.20 *		2. SDis	0.39 *	0.15 *
	3. SDis	0.47 *	0.19 *		3. SDur	0.32 *	0.16 *		3. SL	0.41 *	0.13 *
	4. SQ	0.49 *	0.14 *		4. SL	0.33 *	0.07		4. SQ	0.41 *	0.09
	5. SDur	0.49 *	0.06 *		5. SE	0.33 *	0.03		5. SDur	0.41 *	0.01
	6. SE	0.49 *	−0.02		6. SMed	0.34 *	0.10 *		6. SE	0.41 *	−0.01
	7. SMed	0.49 *	0.05		7. SQ	0.34 *	−0.03		7. SMed	0.41 *	0.01
PHQ-9	1. B-E	0.27 *	0.52 *	GHQ-12	1. B-E	0.20 *	0.45 *	DS	1. B-E	0.16 *	0.40 *
	2. B-RA	0.29 *	0.19 *		2. B-RA	0.23 *	0.20 *		2. B-RA	0.17 *	0.14 *
	3. B-D	0.29 *	0.02		3. B-D	0.23 *	−0.06		3. B-D	0.17 *	0.02
PHQ-9	1. S-A	0.28 *	0.32 *	GHQ-12	1. S-F	0.30 *	0.55 *	DS	1. S-F	0.30 *	0.55 *
	2. S-F	0.31 *	0.26 *		2. S-A	0.33 *	0.27 *		2. S-A	0.33 *	0.23 *
	3. S-PS	0.31 *	0.02		3. S-PS	0.33 *	−0.03		3. S-PS	0.33 *	0.02

DV = dependent variable; PC1-7 = stressor categories; SDur = sleep duration; SQ = sleep quality; SL = sleep latency; SE = sleep efficiency; SDis = sleep disturbance; DDys = daytime dysfunction; SMed = sleep medication; B-E = burnout exhaustion; B-RA = reduced sense of accomplishment; B-D = sports devaluation; SP-F = social phobia anxiety fear; SP-A = social phobia anxiety avoidance; SP-PS = social phobia anxiety physiological symptoms; DS = distress screener; * *p* < 0.05.

## Data Availability

The data presented in this study are available on request from the corresponding author. The data are not publicly available due to participants’ privacy protection.

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
