# Peer review of "Examining the Predictors of Mental Ill Health in Esport Competitors"

_healthcare, 2022, doi:10.3390/healthcare10040626_

Round 1

Reviewer 1 Report

Comments to the authors

The reviewer examined the submitted paper titled “Examining the predictors of mental ill health in esport competitors” with great interest. He believes that the paper is well written and impressive. However, at the same time, he has serious concerns regarding the theoretical and methodological aspects of the study.

First, the reviewer wonders as to why the authors focused on the mental ill health of esports athletes. To clarify the reason why the problem of mental illness in esports athletes is significant for academic research, he believes that the authors should explain the uniqueness of esports athletes in comparison to the other people. For example, he thinks that the authors could have brought out the differences in professional characteristics between esports athletes and other sports athletes. Otherwise, the author might have to explain the differences in characteristics between esports athletes and amateur game players. By doing so, the authors revealed the significance of this paper in the field of health studies. As the authors did not bring out the uniqueness of the mental health of esports athletes in the field of health studies, the reviewer thinks that the contribution of this study to that study is unclear. To avoid this problem, the authors should discuss the differences between the mental health of esports athletes and that of other people in the Introduction section by referring to the previous studies. Additionally, they should discuss the causes of stressors threatening the mental health of esports athletes: how different they are when compared to the stressors experienced by other sports athletes or amateur game players.

Second, the reviewer felt that the paper was neither theoretical nor descriptive. As the paper did not specifically explain how stress is generated by playing esports as an elite level player, the mechanism generating mental ill health among esports athletes remains unclear. Certainly, the authors seem to adopt the following scheme to explain the processes of generating mental illness in esports athletes: esports players’ stresses under highly competitive envelopment induce various physical and psychological symptoms, and subsequently cause mental illness. However, this study did not directly examine the causal processes of stress, physical and psychological symptoms, or mental illness. This study only confirmed the correlation between these variables based on cross-sectional data. Therefore, the reviewer does not think that the paper theoretically clarifies the causal mechanism of mental illness in esports athletes.

Third, the reviewer believes that the study did not have practical implications. The effectiveness of the intervention could be justified only when the causal relationships between the factors were clarified. However, as mentioned above, the study failed to specify the causal mechanism generating mental illness in esports athletes. Therefore, it cannot be said that the authors succeeded in proving the effectiveness of the intervention in the environment surrounding the esports athletes.

Last, the reviewer feels that the tables shown in the paper have low readability because of them having too many variables and these variables being highly complex. He wondered why the authors adopted hierarchical regression models instead of structural equation models. By adopting structural equation models, the authors could have efficiently extracted the latent factors, which would have enabled them to analyze the mental health of esports athletes with a simpler framework. If the authors adopt hierarchical regression models, the reviewer believes that the authors should present the merits of using hierarchical regression models in the study.

Author Response

Please see the attachment for detailed responses to the reviewer's feedback

Reviewer 2 Report

The paper aims to describe the relationship between stressors resulting from competing in esports, other predictors (e.g., sleep, burnout, social phobia), and some mental health symptoms dimensions. In the study, the authors use a transactional study with inferential statistics to predict the relationship between stress, other predictors, and reported symptoms.

Some necessary precisions and corrections might help to achieve a contribution.

Mainly, authors could briefly define esport athletes´ activities (what is counter-strike: Global Offensive, Valorant, and Rainbow Six Siege?) and their individual esports story (e.g., how long they have been competing or involved in such activities, and how many hours a day apply on it).

The authors might also clarify if they consider mental health measured by anxiety or depression instruments. Have they been the same variable? Sometimes they referred to mental health, and others like depression or anxiety (e.g., lines 58 – 60: “mental ill-health… have been seen to predict low mood…). Is Social phobia anxiety a predictor of mental health illness?

The authors might consider basing the affirmation from lines 39 to 41 in some systematic research papers.

The authors might define what CS:GO refers to in line 50.

The kind of study, number of participants, and lack of some instrument’s validity information limit the level of the paper´s aim defined as the predictive relationship between named predictors, stressors, and mental health illness.

The authors proposed a predictive relationship between stressors and mental health symptoms mediated by sleep quality-burnout-social phobia through establishing the five hypotheses. However, the recommendation is to describe associations once instruments´ validity data be incorporated: PSQI, ABQ, SPIN, GHQ-12, PHQ-9, and 4DSQ rather than the hypothesis about predictions, or use path analysis models to explore such mediated predictive relationships.

Thus, the information about the validity of PSQI, ABQ, SPIN, GHQ-12, PHQ-9, and 4DSQ, is needed to assure the distribution of the factors considered through means dimensions/scales calculations (e.g., subjective sleep quality). Consider adding the range of each instrument´s punctuation (minimum-maximum) on the measures apart.

Once the authors consider the validity information of the instruments, Pearson´s correlations provide data about independence between dimensions. However, regression analysis, considered per relationship, increases the error type 1 probability. The authors might consider getting a path analysis model fit to describe the level of the relationship since they thought in mediated predictors routs.

The authors also might consider what they are measuring, the stress of competing at an elite level, or the stressful situations related to the esport activities (e.g., line 269-270 vs. line 275).

The validity of the instruments, and the fit indices models showing the level of measurement and predictability through mediated effects, respectively, might result in the certainty of the conclusions. Results not jet offer strong support about the predictive relationships.

Author Response

(The authors gave the same response as above.)

Reviewer 3 Report

The paper by Smith et al. analyzed eSports athletes mental health and correlates with stressors, sleep, burnout, social phobia anxiety. To this end, the stressors, sleep, burnout, social phobia anxiety and mental ill health were described. The authors tested the subjects using a questionnaire developed for the study purpose, from which seven factors describing stressors pertinent to eSports athletes were derived for analysis. Furthermore, they also tested burnout, distress, general anxiety and depressive symptoms and depression, sleep troubles and social anxiety. The title and abstract are appropriate for the content of the text. Furthermore, the article is well constructed, the data analysis and statistics were well conducted, and analysis was well performed.

Author Response

We thank the reviewer for the positive comments about our paper. We are very happy that you see value in the purpose of the study, and feel like the statistical analyses conducted were appropriate to the study and performed well.

Reviewer 4 Report

The article reflects research into a relevant and under-researched topic,  in an occupational field that is not strictly professional. It presents structure and content strengths that make it worthy of consideration for publication. The empirical analysis is conducted with data from a restricted sample but allows the exploratory approach presented in the results discussion and conclusions.

Author Response

We thank the reviewer for the positive comments about our paper.

Round 2

Reviewer 1 Report

The reviewer examined the revised manuscript carefully.

Consequently, he confirmed that his concerns had been addressed adequately.

He recommends the editor to publish the manuscript on Healthcare.

Author Response

We thank the reviewer for their feedback in the review process and we are pleased that they are happy our article be published